# Indoor and Outdoor Design in Healthcare Environments: The Employees' Views in the General University Hospital of Alexandroupolis, Greece

**Paraskevi Karanikola** [1] , **Veronika Andrea** [1,*] , **Stilianos Tampakis** [2] **and Anastasia Tsolakidou** [1]

[1]   Department of Forestry and Management of the Environment and Natural Resources, Democritus University of Thrace, 193 Pantazidou Str. Orestiada, 68200 Orestiada, Greece; pkaranik@fmenr.duth.gr (P.K.); zikidis@hotmail.com (A.T.)

[2]   Faculty of Forestry and Natural Environment, School of Agriculture, Forestry and Natural Environment, Aristotle University of Thessaloniki, 54124 Thessaloniki, Greece; stampaki@for.auth.gr

*   Correspondence: vandrea@fmenr.duth.gr; Tel.: +30-25520-41136

**Abstract:** Healthcare environments should be designed and operate as healing places for all their users. Therefore, the design of outdoor and indoor spaces, has to be oriented towards distressing solutions. The employees' occupational stress affects their feelings and in turn their services they provide. Thus, this study aimed at the evaluation of the General University Hospital of Alexandroupolis, Greece according to its employees' views. With the use of two step cluster analysis and the hierarchical cluster analysis, important findings were derived, concerning the interior and landscape design of the healthcare environment. The hospital indoor and outdoor spaces were investigated in relation with environmental parameters and psychological effects on their users. The results have shown a lack of the appropriate green spaces—even though their beneficial role was acknowledged—and marginal satisfaction with available spaces. Conclusively, it should be noted that there is still room for improvements in both interior and outdoor premises of the hospital to reduce stress levels, especially for its nursing staff.

**Keywords:** bioclimatic architecture; outdoor; green spaces; indoor; healthcare environments; Alexandroupolis; hospital; occupational stress

## 1. Introduction

Hospitals are healthcare environments that should be designed under special conditions in order to create a sense of comfort and significantly reduce stress levels of patients, employees and of all their users. Already, since the last decade, several surveys had investigated the role of indoor and outdoor spaces in health care units; especially, as regards the patients' or the staff's safety and their stress relief potential [1,2]. While, on current times, the overgrowing pressure for health care services presupposes a better understanding on comfort and physical settings in healthcare facilities [3] in order to meet their users' needs.

Direct and indirect contact with nature could serve as a beneficial asset in designing hospital spaces. Some of the wholesome impacts of green spaces, which appoint them as an imperative when designing and operating healthcare facilities, include health benefits; recreational or aesthetic values; escape for stressed employees; development of public relations; and air quality improvement [4]. Moreover, exposure to green spaces has turned to be salutary for mental illness recovery [5], not to mention that the introduction of items deriving from the natural environment in hospital settings are closely affiliated with advanced healing and treatment outcomes, as well as with the patient's compliance with rehabilitation therapies [6]. The existence of green indoor and outdoor spaces incorporation of

indict environmental components—indoors, such as landscape window view—are proven to create a positive atmosphere for hospitalized children and their companions in Iran [7].

Added to that, and due to recent findings, design efficacy in healthcare environments were identified as an important factor for the high levels of distress observed at the patients' visitors. Direct exposure to nature through the creation of gardens would be an indicate innervation for outdoor space and landscape design. In fact, Ulrich et al. [8] found that patients' family members that used the garden rather than indoor spaces as a way out from stress, have managed to deal more efficiently with depression feelings.

Yet, not only patients or visitors experience stress in hospitals, but also staff members. In fact, the latter are characterized by occupational stress and burnout as they spend a lot of hours in these special and very demanding environments during a hectic workweek. Environmental components, either direct or indirect ones, hold a significant role in reducing work stress for employees. In particular, they are regarded as of utmost importance in facilitating activity and promoting better mental health and well-being [9]. However, it should be underlined that they comprise a cost-effective intervention for creating well-structured and positive health care environments. In fact, their overall contribution on public health has gained increasing attention over the last decades [10].

Staff efficiency was found to fruitfully interact with healthcare environments that were carefully and evidence-based designed with regard to their physical settings [2]. Stressful feelings could be alleviated and help hospital workers improve their provided healthcare services if they actively interact with natural landscapes, plants or a garden [11]. In a staff-oriented approach, Applebaum et al. [12] state that work environment is closely related with occupational stress for nursing staff in healthcare units. Actually, they argue that anxiety feelings have an impact on their job satisfaction and on their intention to change occupation. The same were led to very important inferences and correlations among environmental factors such as odor, noise, light, color; work stress and satisfaction with work; which in turn proved that workplace environment holds a critical position in mitigating work stress. In many cases in the USA it was also evident that nurses were exposed in high degrees of occupational stress [13,14], which inevitably leads to an overgrowing shortage of registered nurses [15] and rising numbers in nurse retirement [16,17]. It should be noted that occupational stress in healthcare environments embeds risks for lower performance levels and medical errors while, it poses threats for over burnout and attrition [2,18].

However, in many cases, the spatial and comfort methodologies used in design and construction of healthcare spaces have followed the structures and disciplines used in offices [12]. It was not conceptualized that healthcare outdoor and indoor spaces ought to provide a therapeutic environment. For instance, although it is acknowledged that fostering physical environments in healthcare sector could enhance positive feelings for all users, many hospital spaces are windowless or substantially occupy basements as workplaces for staff and for patient handling [6]. However, these kinds of settings are not able to provide a relaxing "green" view to its users, whose views are important in hospital outdoor and indoor design, as their understanding will assist managers and decision makers to meet the users' needs [19].

It should be also noted that proper design in outdoor and indoor spaces should be focused on bioclimatic architecture design to achieve energy efficiency goals. According to Watson [20], bioclimatic design constitutes a major green perspective of architecture that combines "biology and climate" by designing indoor and outdoor spaces in line with the existing local climatic conditions. He interestingly notes that the reference of the word "bioclimatic" implies the linkage of architectural design to the physiological and psychological necessity for achieving good health and comfort. The adaptation of bioclimatic principles presupposes that architects aim to create comfort by the integration of the existing microclimate. Specific strategies are designed and implemented in order to ensure energy efficiency by the utilization of the local prevailing conditions in terms of temperature, humidity, solar radiation and winds [21].

More specifically, the utilization of plants such as native trees and shrubs, would offer natural shading and improve passive cooling and ventilation of indoor and outdoor spaces [22]. Adopting green solutions would improve the aesthetic value of indoor and outdoor spaces and at the same would enhance the environmental performance of buildings. The transformation into the so-called "green hospitals" is an era that receives a lot of attention nowadays. Decision makers and designers aspire to achieve sustainability in the construction and operation of healthcare units by the adoption of green practices in outdoor and indoor design. The improvement of accessibility, energy and water use efficiency, and the adoption of sustainable measures and disciplines in the design of spaces are considered as components of green hospitals and green health services, able to perceive wellness for healthcare unit users [23]. Thus, hospital spatial design demands an integrated plan for achieving healing purposes, enhance the patience experience [24] and provide a better workplace for employees. Its contribution to employees' performance and well-being is another structural component in efficient outdoor and indoor planning supported by the concept that healthcare units should not be considered only as buildings for treatments [22].

The case study was focused on the evaluation and improvement of the premises comprising the General University Hospital of Alexandroupolis, Greece. To this end, it was attempted to investigate and analyze—from a sociological aspect—the employees' views and correlations concerning the impact of the hospital design, and particularly addressing indoor and outdoor spaces. In line with the existing settings, direct and indirect contact with the natural environment was also examined in this study. Indicative results reveal that there is a deficiency in green spaces, although the hospital staff recognize their beneficial role in improving the hospital as a workplace. Specific solutions were proposed addressing interior and landscape design eligible for healthcare environments with similar characteristics, such as the creation of a garden, the placement of natural flowers or plants, and encouragement of taking care of them. The suggested guidelines are based on bioclimatic architecture, landscape architecture, the existing climate and environmental conditions, and also by taking into consideration that nursing staff falls to high levels of occupational stress.

## 2. Materials and Methods

### 2.1. Study Area

The study area was the General University Hospital of Alexandroupolis (GUHA). GUHA is a state hospital, which belongs to the National Health System of Greece. Since December 2002, it is a health care provider situated in Dragana, an area situated 6km away from the city of Alexandroupolis (Figure 1). Its building complex has a total area of 93,544 m$^2$ and is built on a plot of 200,000 m$^2$. There are 673 beds available at the hospital. GUHA is characterized by administrative and financial independence, as it is a decentralized and independent health care unit of the 4th Health District of Macedonia and Thrace. The unit is supervised by the Ministry of Health. The city of Alexandroupolis is situated in the northern part of Greece and it is the capital of the Regional Unit of Evros. This Regional Unit of Greece borders with Bulgaria to the north and with Turkey to the east. The Regional Unit of Evros administratively belongs to the Region of Macedonia and Thrace.

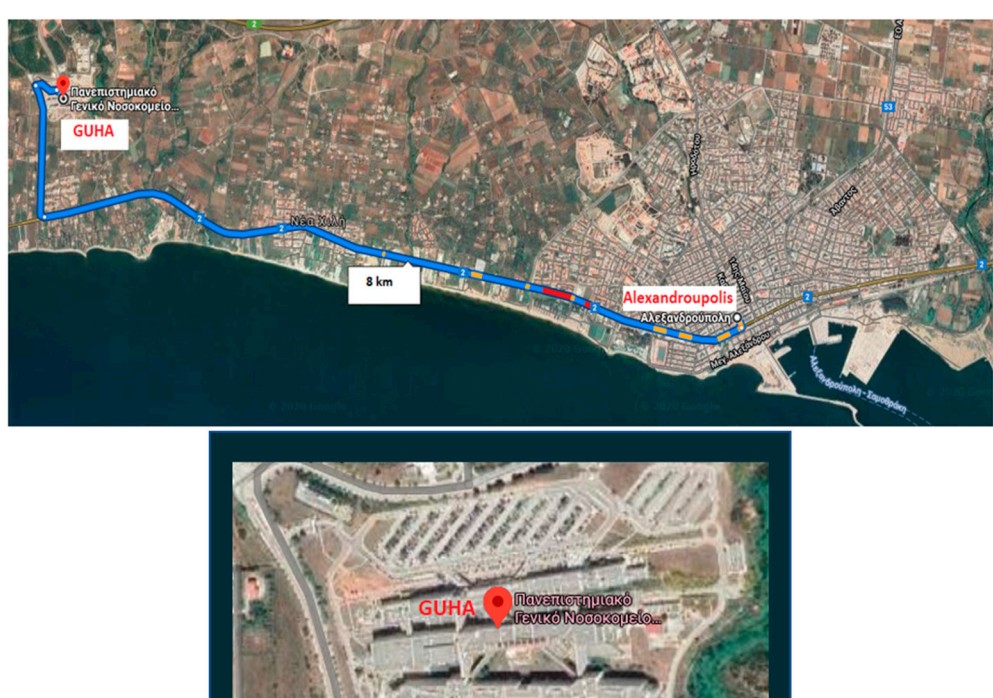

**Figure 1.** The location of the GUHA and its distance map from the city of Alexandroupolis, Greece. (Source: Google Maps).

## 2.2. The Survey

The data were collected in 2018 with the aim of personal interviews. The interview is a common method for the collection of statistical data while, it is often used in sampling research [25]. The population under study included all the employees occupied in the GUHA. The employees of the hospital were divided into four groups:

1.  The medical staff consisting of 370 people;
2.  The nursing staff of 563 people;
3.  The administration staff of 80 people;
4.  The staff from other departments such as paramedical, technical, scientific non-medical and others consisting of 187 people.

The survey was conducted with the official permission of the hospital administration. However, it should be noted that certain restrictions were set in the concept that a hospital constitutes a working place of special working conditions. Therefore, it became possible to collect 268 questionnaires depicting the 22.33% of the GUHA employees. The survey was divided into five sections:

1.  The employees' demographics and profile;
2.  The impact of green spaces on the improvement of the working healthcare environment;
3.  Evaluation of the existing GUHA outdoor environment;
4.  Evaluation of the existing GUHA indoor environment;
5.  Two Step Custer Analysis correlations on indoor—outdoor environment and demographics.

For the data collection, face-to-face interviews took place by means of a combination of close-ended questions and Likert-scale questions. The questionnaires included a significant range of topics in order

to investigate the existing outdoor–indoor design; the green spaces; and the employees' personal commitment to take care of the; accessibility; bioclimatic design architecture and measures; raising and awareness on energy efficiency; and sustainability in buildings.

### 2.3. Research Method

In order to collect the data, the employees' participation in the survey was critical. Yet, the approaching process—in order for the employees to complete the questionnaire in a workplace such as a hospital—is distinguished by high complexity and a challenging nature. This could be explained through the fact that most of the hospital staff have to follow a very demanding working program. In fact, this program requires that they are occupied within three shifts that take place 24 h a day for the course of a week. While, the staff can take their days off in between, under an unstable week schedule. The use of a nominal catalogue and the conduction of sampling was rejected, as it would not ensure the anonymity of the respondents. Hence, 300 questionnaires were distributed randomly, and then the respondents were asked to place them into a ballot box. With the implementation of this method for the data collection, finally a percentage of 22.33% employees participated in the survey. In addition, the random selection of the respondents and the low levels of responding refusal (approximately 1%)—mainly due to the hectic workload—led to the acceptance of the condition that the sample was representative for total of the GUHA employees. The data collection took place in 2018 and the statistical package SPSS was used for the data processing. In the multivariables "*factors characterizing the outdoor environment of the GUHA*" and "*factors characterizing the indoor environment of the GUHA*", reliability and factor analysis were applied [26,27]. Aiming to investigate the internal reliability of a questionnaire [28], i.e., if our data had the tendency to measure the same thing, we used the coefficient (or Cronbach's reliability coefficient). A coefficient equal to or higher than 0.70 is considered as satisfactory [29], while for values higher than 0.80 it is considered as very satisfactory. In practice, reliability coefficients with values lower than 0.60 were also accepted in many cases [26]. The tests must be reliable in order to be useful. In fact, reliability does not suffice; it should also be valid, and this is checked through the implementation of factor analysis [26].

Factor analysis is a statistical method which aims to discover the existence of common factors within a group of variables [26]. More specifically, principal component analysis was used in the case study. The selection of the number of factors is a dynamic process and presupposes the evaluation of the model in a repeated fashion. To this end, the criterion of the smooth slope was used on the scree plot [30]. The rotation of the matrix principal components was implemented by the use of the maximum variance rotation method by Kaiser [31].

The hierarchical cluster analysis was applied to examine data sets comprising multiple variables and to identify possible groupings of the data [32]. In fact, it can be operative both towards the direction of the observations' grouping, and to the direction of the variables' grouping [26]. The Pearson correlation coefficient was used as a distance measure, and the method of the nearest neighbor was applied to combine the cluster observations.

Eventually, for the statistical clustering of the employees into given distinct groups–Two Step Cluster Analysis was selected to further process clusters that were produced by the factor analysis. This analysis serves as an exploratory tool for the identification of clusters that include similar objects within a large number of observations. Taking as granted that the variables are independent of each other, it allows the manipulation of categorical and continuous variables at the same time. In addition, with the implementation of Pearson's $X^2$ check, the relationship between other variables and each cluster was investigated separately. In this way, the identity of each cluster was determined accurately.

## 3. Results and Discussion

### 3.1. The Employees' Demographics and Profile

The group under investigation included the employees of the GUHA in Greece. Their demographic characteristics were recorded and the findings reveal that 65.7% of the respondents are women and 34.3% are men. Concerning their age, 47% of respondents are aged between 41 and 50, the 22.4% in the age class of 31–40, 22.4% are over 50 years old and 8.2% belong in the age class of 18–30.

As for their working experience, it was evident that 28.4% are occupied in the health care unit for 10.1–20 years, 27.2% for up to 5 years, 22% have been working at the GUHA for 5.1–10 years, and the 19% for more than 20 years. This fact shows that professional recruitment and development in healthcare environments takes place at an old age in Greece.

Regarding their marital status, employees are mainly married (69.8%), while 24.6% are unmarried and 5.6% are divorced or widowed. In addition, almost 4 out of 10 (42.9%) have 2 children, 32.8% have no children, 15.7% have one child, 6.7% have 3 children and 1.9% have 4 or more children.

Concerning their education level, it seems that most of them (41.4%) have a high education level and hold a higher education degree (university), and the 32.5% are educated in a technological educational institute. Furthermore, 11.9% are high school graduates and 11.2% are technical school graduates. Lower rankings receive the rest educational degrees such as secondary school education completed by the 1.9% of the respondents while, 1.1% are primary school graduates.

In reference with their working position it was stated that 28% of the employees are occupied as medical staff, 32.5% as nursing staff, 30.6% as administrative staff and 9% in another field of employment.

Their satisfaction with their employment and working position was then investigated. Therefore, it was noticed that the majority (61.2%) are satisfied with their work. In addition, 1.5% are completely satisfied and 7.8% are very satisfied, whilst 25.7% are less satisfied and 3.7% are not at all satisfied.

Moreover, they are proven to be less satisfied with their income (50%), while 15.7% claimed to be not satisfied at all. Nonetheless, 32.5% say that they are satisfied with their income, 1.5% are very satisfied, and a slight minority of 0.4% argue that they are completely satisfied with their incomes.

### 3.2. The Impact of Green Spaces on the Improvement of the Working Healthcare Environment

The hospital, as a workplace, is different from other workplaces. Besides the physical fatigue of employees, the psychological stress from the difficult situations the employees experience in their daily lives, leads them to reach the end of their mental resources, even if their training and daily familiarity with stressful incidents make them stronger and more capable of dealing with them. There are times when they feel they need to escape from the pressure of work-related stress (Table 1). For the respondents, the most important ways out in order to retreat work stress is to gaze at the outdoor environment from the windows and head to the outdoor environment to isolate and calm down. Other attitudes follow, such going at a special place (café) to discuss with friends; going to a special place (toilet) to isolate and smoke a cigarette. Smoking and the purchase of tobacco products are prohibited inside the hospital, but this unhealthy habit is allowed at outdoor spaces.

**Table 1.** Escape—employees' need due to working pressure.

| Important Ways out in Order to Retreat Work Stress | Always | Often | Sometimes | Rarely | Never |
|---|---|---|---|---|---|
| They look through the windows the outdoor environment | 8.2% | 29.9% | 40.7% | 13.8% | 7.5% |
| They go to the outdoor environment to isolate and calm down | 4.5% | 20.9% | 31.7% | 23.1% | 19.8% |
| They go to a special space to discuss with friends (café) | 1.9% | 13.8% | 28.7% | 34.3% | 21.3% |
| They go to a special place to isolate (toilets) | 2.2% | 4.5% | 19.0% | 33.6% | 40.7% |
| They smoke a cigarette | 4.5% | 13.8% | 13.4% | 11.9% | 56.3% |

Additionally, during the staff's shift, the visitors of the patients, who gather in the hospital, are claimed mainly to disturb them (44%), or leave them indifferent (42.5%), while 2.2% of the employees have stated that they have fun and 10.8% reported something else.

To this end it should be noted that absence of green spaces from the anthropogenic environment has negative effects on all aspects of human activity. Humans are affected psychologically, spiritually, culturally, and also as regards their workplace environment [33]. The above problem is exacerbated in a place that is by nature stressful and depressing, like the one of a hospital. Under this concept it is highly suggested the creation for a well-designed hospital garden. This green space will serve as a mean for the promotion of safety and the relief from stress and pressure [23]. Not to mention that it will help improving social contacts and socializing activity in general [34]. At the same time, it will enable people who visit the healthcare environment to enjoy nature and develop their senses [35]. Thus, the creation of green outdoor spaces in healthcare environments is considered as beneficial for health promotion [36] and very necessary from an aesthetic point of view [9].

In the same basis, the impact of green spaces, such as the creation of a garden in the GUHA, might be significant for the improvement of its users' mental health. It could serve also as a structural component for stress relief, caused by work pressure, and for the reduction of odds of depression as they are positively associated with perceived mental health [37]. Respectively, the existence and care of plants in workplace will assist in reducing pressure, created by the nature of work in the hospital. To this end the employees were asked about their relationship with plants in the hospital. In particular, 36.2% reported that they rarely have plants in the workplace, 26.1% never, 24.6% sometimes, 8.2% often, and 4.9% of the employees' stated that they always have plants in GUHA. Respectively, 58.2% of the hospital staff stated that they do not take care of plants in their workplace; while, the 29.5% claims to take care of them sometimes and 12.3% of the questioned noted that usually take care of them.

*3.3. Evaluation of the Existing GUHA Outdoor Environment*

The GUHA employees were asked to evaluate the existing outdoor environment. Thus, 43.7% of the hospital staff argued that they are satisfied with the hospital outdoor environment, 42.5% claim they are less satisfied, 6.3% are very satisfied, 5.6% are not at all satisfied. Additionally, the 1.9% are completely satisfied. The corresponding results that emerged from a predate thesis implemented in 2003 by Anthopoulos [38], have shown that 12.7% were then very satisfied with the existing outdoor environment in 2003. The majority (87.3%) though have stated that they were not at all or less satisfied. The new findings revealed that important improvements have taken place in the surrounding outdoor facilities of the GUHA, as almost half of the respondents are now satisfied or very satisfied with the current conditions.

As green spaces constitute a significant feature of the outdoor environment, the respondents were called to assess them. In particular, 46.3% regards them as mediocre, 27.2% as poor, 17.2% as good, 7.8% as very poor and 1.5% as very good. According to the above mentioned, it is clear that there are many prospects for the improvement of green spaces in the GUHA.

Another point to consider is that the GUHA is located in an area surrounded by the sea. On this ground, it can be assumed that it is an acceptable place for a permanent residence. Thus, 10.8% of the hospital staff considered the prospect of settling permanently close to the hospital as very good, and the 42.2% as good, while 28.7% of the hospital workers believe that it is a moderate place for permanent establishment, 13.1% of the staff believe is a poor place and 5.2% access it as very poor.

Accessibility provisions were also analyzed in this study. More specifically, the 44.4% of the GUHA employees regard that they are satisfied with the accessibility provisions of the hospital, 22.8% say they are very satisfied, 17.5% are slightly satisfied, 12.3% are completely satisfied and 3% are not at all satisfied. The public transport to GUHA offered in the urban area is organized and provides fixed and frequent routes by bus. This facilitates accessibly by the use of public transportation means. However, the vast majority (79.9%) of the hospital staff declare that they use their car to reach GUHA. Moreover, 2.2% move by motorcycle, 0.7% by bicycle, 0.4% by something else, and only 16.8% use public transport to get to their workplace.

The Implementation of Factor Analysis on Outdoor Evaluation

The results from the evaluation of the existing GUHA outdoor environment, are listed in Figure 2. In this representation the assessment valued with "*1*" shows the lowest satisfaction, while assessment with value "*10*" depicts the highest satisfaction level. The lowest evaluation is received by the variables, provision of playground spaces (2.81); safety for children (3.43); drinking water (3.95); plant diversity (4.05); plant care (4.18); amenities for people with disabilities (4.25); leisure infrastructures (4.25); and the presence of stray or accompanying animals (4.69). The following variables are the architectural landscape design (5.07); the quality of the building materials used for outdoor infrastructures coating (5.27); unpleasant odors (5.39); cleanliness (5.62), the color of the surrounding walls (5.64); and the noise pollution (5.90). In contrast, more positively are evaluated the air quality (6.06); the thermal comfort (6.44); the visual comfort (6.81), the number of users allocated in the space (6.81); and the accessibility to arrive and depart from the GUHA (7.15). Finally, as regards the highest evaluation level, the total occupied space (7.65) and the available parking spaces (7.69) are the prevailing variables in this classification by the employees.

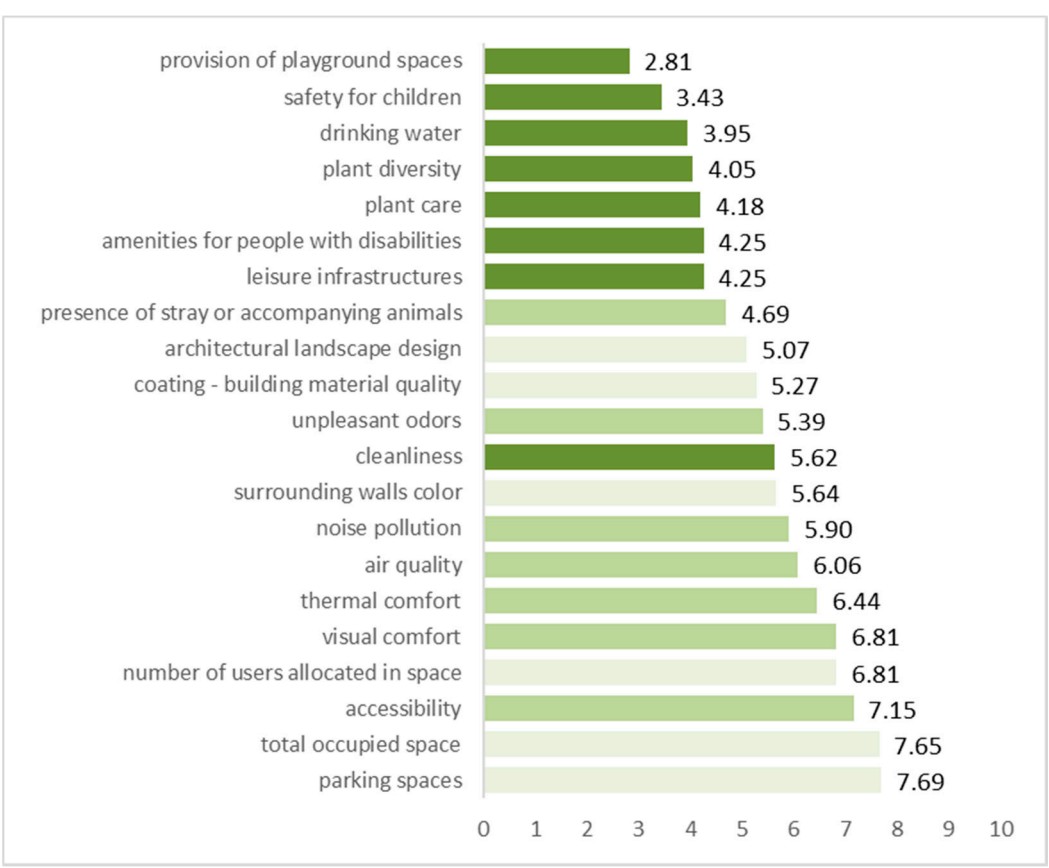

**Figure 2.** Evaluation of the existing GUHA outdoor environment (the different colors in the variables represent the three factors arisen by the factor analysis which follows).

To test the consistency of the equivalent questions of the above multivariables, we used reliability analysis. The value of the coefficient alpha is significantly high (0.922). This is a strong indication that clusters are reasonably consistent, meaning that the data have the tendency to measure the same thing. Prior to the application of factor analysis, we conducted all the necessary checks for the data and variables' appropriateness to be used in the model. Table 2 reveals the loads that are the partial correlation factors of the 21 variables, with each of the three factors resulting from the analysis. The higher load of a variable in a factor, the more this factor is responsible for the total degree of fluctuation (0.5) of the considered variable.

**Table 2.** Factor analysis loadings after rotation (bold numbers show the factor that belongs to each variable).

| Variable | Factor Loadings | | |
|---|---|---|---|
| | **1** | **2** | **3** |
| total occupied space | 0.050 | 0.151 | **0.781** |
| parking spaces | 0.053 | 0.235 | **0.726** |
| coating—building material quality | 0.351 | 0.251 | **0.558** |
| surrounding walls color | 0.439 | 0.128 | **0.549** |
| accessibility | 0.222 | **0.501** | 0.498 |
| number of users allocated in space | 0.166 | 0.402 | **0.614** |
| architectural landscape design | 0.435 | 0.313 | **0.525** |
| leisure infrastructures | **0.755** | 0.107 | 0.354 |
| cleanliness | **0.420** | 0.266 | 0.341 |
| plant diversity | **0.764** | 0.075 | 0.321 |
| plant care | **0.748** | 0.116 | 0.274 |
| provision of playground spaces | **0.837** | 0.144 | −0.072 |
| safety for children | **0.751** | 0.276 | −0.006 |
| amenities for people with disabilities | **0.595** | 0.230 | 0.271 |
| presence of stray or accompanying animals | 0.351 | **0.576** | 0.080 |
| noise pollution | 0.178 | **0.701** | 0.169 |
| unpleasant odors | 0.272 | **0.717** | 0.144 |
| drinking water | **0.438** | 0.310 | 0.038 |
| air quality | 0.025 | **0.729** | 0.304 |
| thermal comfort | 0.097 | **0.670** | 0.302 |
| visual comfort | 0.236 | **0.581** | 0.395 |

The first factor includes the variables leisure infrastructures; cleanliness (with load < 0.5); plant diversity; plant care; provision of playground spaces; safety for children; amenities for people with disabilities; and drinking water (with load < 0.5) and could be termed as "*outdoor infrastructures for recreational usage*". Respectively, the second factor could be titled as "*comfort features of outdoor spaces*"; and consists of the variables: accessibility; presence of stray or accompanying animals; noise pollution; unpleasant odors; air quality; thermal comfort; visual comfort. The third factor was formatted by the variables: total occupied space; parking spaces; coating—building material quality; surrounding walls color; number of users allocated in space; architectural landscape design. While, the variable "accessibility" holds high loadings also in the third factor (0.498), which means that it serves as a bridge between the second and the third factor. The third factor could be named as "*landscape design*".

*3.4. Evaluation of the Existing GUHA Indoor Environment*

In what follows, GUHA employees were asked to evaluate the existing indoor environment. Consequently, 45.1% of the employees affirm less satisfied with the interior of the hospital, 44.8% are said to be satisfied, 7.1% not at all satisfied, 3% very satisfied, while no one stated to be completely satisfied with the existing indoor environment of the GUHA.

The GUHA was then evaluated as a workplace via its staff satisfaction level. Namely, 47.8% of the hospital staff believe that the health care unit of Alexandroupolis is a moderate workplace, 33.2% regard it as good, 2.2% as very good, 11.6% as poor and a percentage of 5.2% as very poor.

The Implementation of Factor Analysis on Indoor Evaluation

The results from the evaluation of the existing GUHA indoor environment are listed in Figure 3. In this representation, the assessment with "*1*" shows the lowest satisfaction, and it reaches the assessment with value "*10*" depicting the highest satisfaction level. The lowest evaluation is received by the variables, drinking water (4.03); the quality of the building materials used for indoor infrastructures coating (4.09); accessibility among the building floors (4.24); informational signs (4.50); safety for

children (4.63); amenities for people with disabilities (4.74); architectural design (4.75); and color of walls (4.99). The variables that follow are the unpleasant odors (5.01); the air quality (5.25); the noise pollution (5.70); available waiting rooms (5.76); artificial lighting (5.78); natural light (5.87); accessibility in the same floor (5.90); cleanliness (5.97); and the number of users allocated in the space (5.99). The rest of the indoor features are assessed in a more positive light, such as the thermal comfort (6.19), the visual comfort (6.45), the restaurants (6.82) as well as the total occupied space by the indoor complex (6.90).

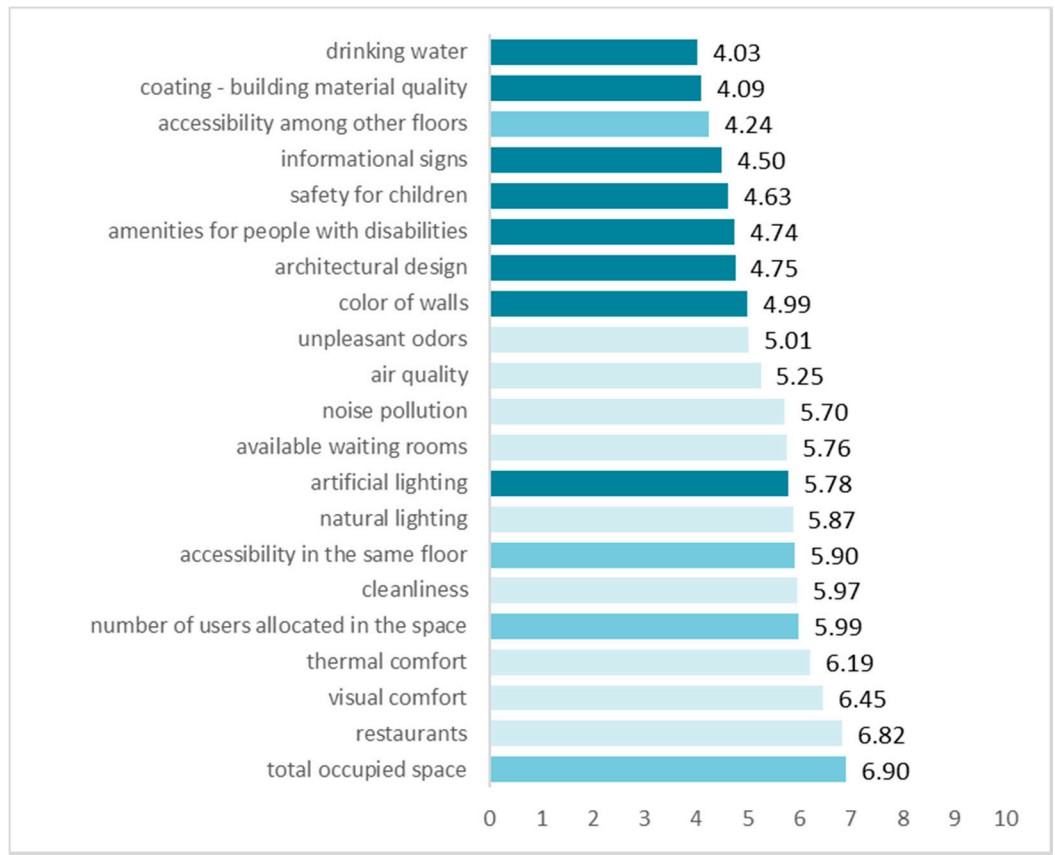

**Figure 3.** Evaluation of the existing GUHA indoor environment (the different colors in the variables represent the three factors arisen by the factor analysis which follows).

To test the consistency of the equivalent questions concerning the above multivariables, we used reliability analysis. The value of the coefficient alpha is significantly high (0.943). This is a strong indication that clusters are reasonably consistent, meaning that the data have the tendency to measure the same thing. Before proceeding with the application of factor analysis, we conducted all the necessary checks for the data and variables' appropriateness to be used in the model. Table 3 reveals the loads that are the partial correlation factors of the 21 variables, with each of the three factors resulting from the analysis. The higher the load of a variable in a factor, the more this factor is responsible for the total degree of fluctuation (0.5) of the considered variable.

The first factor could be named as "*comfort features of indoor spaces*" and includes the variables cleanliness; available waiting rooms; restaurants; natural lighting; noise pollution; unpleasant odors; air quality; thermal comfort; and visual comfort.

**Table 3.** Factor analysis loadings after rotation (bold numbers show the factor that belongs to each variable).

| Variable | Factor Loadings | | |
|---|---|---|---|
| | **1** | **2** | **3** |
| total occupied space | 0.185 | 0.060 | **0.804** |
| accessibility in the same floor | 0.312 | 0.311 | **0.755** |
| accessibility among other floors | 0.161 | **0.577** | 0.522 |
| coating - building material quality | 0.073 | **0.725** | 0.336 |
| color of walls | 0.295 | **0.569** | 0.405 |
| number of users allocated in the space | 0.405 | 0.398 | **0.494** |
| architectural design | 0.343 | **0.546** | 0.453 |
| cleanliness | **0.505** | 0.291 | 0.403 |
| available waiting rooms | **0.566** | 0.434 | 0.281 |
| restaurants | **0.545** | 0.123 | 0.301 |
| safety for children | 0.383 | **0.551** | 0.158 |
| amenities for people with disabilities | 0.316 | **0.642** | 0.275 |
| informational signs | 0.126 | **0.678** | 0.102 |
| natural lighting | **0.660** | 0.375 | 0.194 |
| artificial lighting | 0.483 | **0.568** | 0.267 |
| noise pollution | **0.724** | 0.258 | 0.189 |
| unpleasant odors | **0.655** | 0.460 | 0.125 |
| drinking water | 0.386 | **0.668** | −0.154 |
| air quality | **0.720** | 0.350 | 0.089 |
| thermal comfort | **0.716** | 0.120 | 0.171 |
| visual comfort | **0.833** | 0.088 | 0.226 |

As for the second factor, which could be termed as *"infrastructures for indoor usage"*. It includes the variables accessibility among other floors; coating—building material quality; color of walls; architectural design; safety for children; amenities for people with disabilities; artificial lighting and drinking water. It should be noted that the variable artificial lighting holds high loadings also in the first factor (0.483). In the same line, the variable unpleasant odors, also holds high loadings in the second factor. Thus, the two variables serve as a bridge between the first and the second factor.

Eventually, the third factor titled as "spatial architecture for indoor design" consists of the variables total occupied space; accessibility in the same floor; and number of users allocated in the space. The variables accessibility among other floors and architectural design receive high loadings also at the third factor, with respective values of 0.522 and 0.453. This leads to the conclusion that they function as a bridge between the two factors.

*3.5. Hierarchical Cluster Analysis and Two Step Custer Analysis Correlations on Indoor–Outdoor Environment and Demographics*

The findings of the hierarchical cluster analysis concerning the factors obtained from the above two-factor analyses are interpreted with the use of the dendrogram of the variables. In this analysis, the factors are referred as variables (Figure 4). The variable "comfort features of outdoor spaces" is very closely related to "comfort features of indoor spaces", creating the first cluster that can be described as "comfort features". Respectively, the variable "outdoor infrastructures for recreational usage" is closely affiliated to the "infrastructures for indoor usage", and comprise the second cluster that can be termed as "usage infrastructure". Finally, the variable "landscape design" is connected at a slightly greater distance with "spatial architecture for indoor design". These two variables are the reason for the formation of the third cluster that can be named as "spatial planning". Thus, based on the findings of hierarchical cluster analysis, it became apparent that employees of GUHA have adopted a specific concept by which they connect and relate similar outdoor and indoor spaces of the hospital.

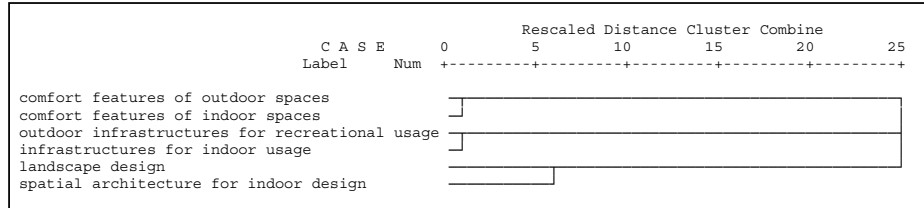

**Figure 4.** The dendrogram of the hierarchical cluster analysis.

The implementation of the two-step cluster analysis took place next and, from this analysis, the observations were classified into three clusters as the best solution. In particular, out of the 240 respondents, 34.6% belong to the first cluster, 30% to the second cluster and 35.4% to the third cluster. Regarding the relative importance of variables in the formation of clusters, the illustrations on Figure 5 diagrammatically provide the statistical significance tests. Variables are important in creating the cluster when the statistical value exceeds the critical value. In particular, it is found that the continuous variables "comfort features of outdoor spaces", "comfort features of indoor spaces", and "spatial architecture for indoor design", are the reasons for the creation of the first cluster. On the same basis, as regards the formation of the first and second cluster, the significant variables are "outdoor infrastructures for recreational usage", "comfort features of indoor spaces", "infrastructures for indoor usage" and "comfort features of indoor spaces". Finally, the variables "outdoor infrastructures for recreational usage", "comfort features of outdoor spaces", "comfort features of indoor spaces", "infrastructures for indoor usage", "spatial architecture for indoor design" and marginally the variable "landscape design", approach the limits of the critical value, and become the reasons for the formation of the third cluster.

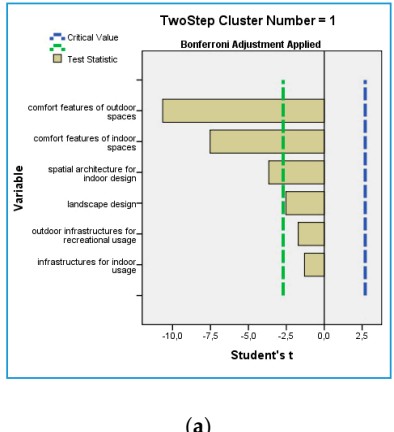

(**a**)

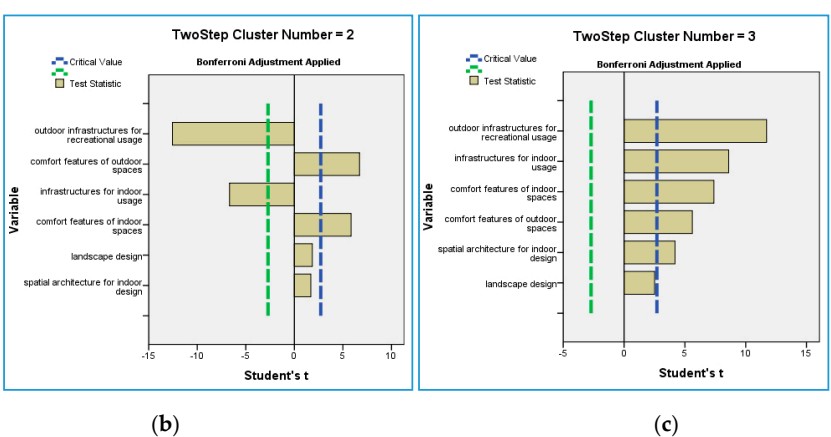

(**b**)                      (**c**)

**Figure 5.** Diagrammatic representations of statistical tests of variables per cluster (**a–c**).

The characteristics of the clusters are listed in Table 4. In particular, the first cluster is characterized by the lowest satisfaction of employees with the outdoor infrastructure for recreational usage, the comfort features of indoor and outdoor spaces, the spatial architecture for indoor design and the landscape design. It is also obvious that in this cluster there is an intermediate or marginal less satisfaction from the existing infrastructure for indoor usage.

**Table 4.** Interpretation of the cluster observations.

| Variables | Cluster 1 | Cluster 2 | Cluster 3 |
|---|---|---|---|
| **Outdoor infrastructures for recreational usage** | less satisfaction | intermediate or marginally less satisfaction | higher satisfaction |
| **Comfort features of outdoor spaces** | less satisfaction | higher satisfaction | higher satisfaction |
| **Landscape design** | less satisfaction | higher satisfaction | higher satisfaction |
| **comfort features of indoor spaces** | less satisfaction | higher satisfaction | higher satisfaction |
| **Infrastructures for indoor usage** | intermediate or marginally less satisfaction | less satisfaction | higher satisfaction |
| **Spatial architecture for indoor design** | less satisfaction | higher satisfaction | higher satisfaction |
| *With the aim of $X^2$ Pearson Test ($\alpha < 0.05$)* | | | |
| **Satisfaction with outdoor spaces** | less or not at all satisfied | less or not at all satisfied | absolutely satisfied to satisfied |
| **Assessment of green spaces** | poor or very poor | mediocre of poor | very good of good |
| **Assessment as permanent residence** | mediocre to very poor | good or poor or very poor | very good of good |
| **Satisfaction with accessibility of the GUHA** | satisfied to not at all satisfied | absolutely to very satisfied | absolutely to very satisfied |
| **Satisfaction with indoor spaces** | less or not at all satisfied | less or not at all satisfied | very satisfied or satisfied |
| **The GUHA as workplace** | mediocre to very poor | mediocre or very poor | very good of good |
| **Education** | technical school or technological educational institute | secondary or technical school | high school or university |
| **Working position** | nursery or other staff | administrative staff | medical staff |
| **Working experience** | 5.1–10 or more than 20 years | 10.1–20 or more than 20 years | up to 5 or 5.1–10 years |
| **Satisfaction with work** | less or not at all or absolutely satisfied | satisfied ι | very satisfied or satisfied |
| **Satisfaction with income** | not at all satisfied | less satisfied | very to less satisfied |

The second cluster is characterized by the employees' highest satisfaction with the comfort features of indoor and outdoor spaces and the spatial architecture for indoor design and landscape design. It is also proven that there is intermediate or marginally less satisfaction as regards the outdoor infrastructure for recreational usage and less satisfaction with infrastructure provided for indoor usage.

The third cluster is characterized by the highest level of satisfaction by the employees with outdoor infrastructures for recreational usage and concerning the infrastructure for indoor usage; the comfort features of indoor and outdoor spaces; and the spatial architecture for indoor design and landscape design.

Indeed, with the aid of Pearson $X^2$ ($\alpha < 0.005$) and in the low part of the Table 4, the correlation among the three clusters with other variables about the employees' characteristics is provided. The representations are the following:

- The employees of the first cluster attribute the most negative evaluation as regards the healthcare environment of GUHA. In fact, they declare that they are less or not at all satisfied with the outdoor spaces of hospital. In addition, they deem that the existent green spaces of the GUHA are poor or very poor. They also evaluate the area, where the hospital is located, as moderate to very poor for permanent residence. While they declare being satisfied or not at all satisfied with the accessibility conditions that are provided in order to reach the hospital. They also argue that they are less or not at all satisfied with the indoor spaces of the hospital. Indeed, they characterize the hospital as a moderate to very poor workplace. This cluster consists of people with an education level of a technical school or a technological educational institute, who are occupied as nursing or other staff. Their working experience varies between 5.1 and 10 or of more than 20 years. These employees are said to be less, not at all, or completely satisfied with their work and not at all satisfied with their income.

- The employees of the second cluster share a moderate view concerning the healthcare environment of GUHA. In general, they show an intermediate evaluation in comparison with responders of cluster one and three. In particular, they are claimed to be less or not at all satisfied with the outdoor spaces of the hospital. However, they estimate that green spaces are mediocre or poor. They also assess the area, where the hospital is located, as good and poor or very poor for a permanent residence. Furthermore, they state that they are absolutely or very satisfied with the accessibility to the hospital. The employees of this cluster feel less or not at all satisfied with the indoor spaces and they characterize the hospital as a mediocre or very poor workplace. They hold an educational degree of high school or technical school. They are occupied in the GUHA as administrative staff, with more than 10.1 years of service. Eventually, the staff comprising the second cluster, also state that they are satisfied with their work, and less satisfied with their income.

- The employees of the third cluster have shared the most positive views in their evaluation about the healthcare unit. They consider themselves as absolutely satisfied to satisfied with the outdoor spaces of the hospital. Moreover, they think that the existing green spaces in the hospital are very good or good. They evaluate the area, where the hospital is located, as very good or good for permanent residence. They believe that they are absolutely or very satisfied with the accessibility conditions of the hospital. They also declare that they are very satisfied or satisfied with the indoor spaces of the GUHA, and they characterize it as a very good or good workplace. This group consists of employees with a high school or university education, who work as medical staff, and they have a working experience of more than 10 years. Finally, this cluster respondents, regard themselves as very satisfied or satisfied with their work, and very to less satisfied with their income.

## 4. Conclusions and Suggestions

The employees of the GUHA are listed in a special category of work staff, as there are executive working conditions established in healthcare environments. This means that, far from the physical pressure due to the tedious nature of work, they also experience significant psychological stress. Thus, there is a need for the staff to have short escapes that will support stress relief processes. In turn, this will assist the improvement of working conditions aiming at the provision of more efficient health services. It should be noted that employees have reported as most important way out from fatigue and occupational stress, the instantaneous images of the outdoor environment captured through windows

and isolation in the outdoor spaces. Looking the landscape out of a window or even a picture of a tree could achieve a built-in, indirect connection with the natural environment. In fact, this is regarded as beneficial for healthcare workers as it positively affects healing process, dealing with stress and enhancement of job satisfaction [39]. Next, it follows that the contact and discussion with friends in a special place (café), isolation in places such as toilets, or smoking a cigarette; although the latter is prohibited in the indoor spaces of hospital.

It was proven that between 2003 and 2018 primary construction or architecture improvements took place in the outdoor environment and infrastructures of the GUHA. This conclusion is based on the more positive evaluation on outdoor environment by the staff of GUHA in this survey, compared with the findings of another study conducted in 2003. Namely, Anthopoulos et al. [40] revealed the disappointment of almost 9 out of 10 respondents concerning the outdoor environment of GUHA in 2003. The same authors have also highlighted the employees' desire for the introduction of therapeutic gardens in the GUHA and the overall improvement of the hospital landscape design. The employees' views are more positive now whereas, according to their evaluation, it seems that there is still room for improvement.

In fact, the better the psychological conditions of the staff, the more these positive feelings are shared among their patients. It is of outmost importance to involve certain practices, such as short breaks, the provision of space to have personal moments with a view of the employees' needs and the encouragement of taking care of plants or animals (such as fish in fishbowls) [39]. Similarly, Cifer and Cifer point out the wholesome effect in stress reduction, by the aesthetic improvement of waiting rooms with the placement of flowers [24]. The incorporation of these solutions for the improvement of the work climate in workplace, should be taken into consideration by the hospital administration, as useful and cost-effective practices. Unfortunately, in GUHA, a small percentage of employees have a plant in their workplace and spend some time to take care of it. However, outdoor direct interaction with components of the natural environment, such as gardening or taking short breaks in a safe garden, could improve the health care environments both as a work and healing place [39]

Under elaborative examination should be also the set of measures addressing the patients' visitors. This is suggested due to the fact that patients' visitors usually put extra pressure on the healthcare staff. In this line, special restrictions could be established on the visitors' behavior concerning time and space allocation in hospital spaces. In fact, the latter is highly suggested, aiming to support the improvement of the staff working conditions.

Nevertheless, the location of the hospital meets the employees' preferences. In fact, it is considered to be situated in an area, where more than half of employees regard it as an acceptable option for permanent residence. However, there is a satisfactory accessibility by public means of transport. On the one hand, almost half of the respondents assess the green spaces of GUHA as mediocre, and the hospital itself as a mediocre workplace. On the other hand, it is declared that indoor and outdoor spaces are reaching satisfactory and merely satisfactory levels.

Evaluating the GUHA indoor and outdoor spaces according to the employees' views leads to the conclusion that existing infrastructures should be improved. Particularly, evaluation on their usage receives low acceptance values, meaning that they have construction failures of damages caused by time, use or weathering. Though features that are closely affiliated with comfort and spatial planning of indoor and outdoor spaces are evaluated under a more positive light. Indeed, indoor and outdoor infrastructures were not listed as a priority by the administration, during the design and operation of the GUHA. This could be explained by the fact that outdoor leisure and playground infrastructures are not regarded as satisfactory facilities. The low prioritization of the proper maintenance and development of such facilities could be attributed to the concept that the presence of children is regarded as a problem for the hospital staff. They probably think that children should not be there, especially when there is no health reason, i.e., as visitors. However, it is a fact that children accompany their parents on patients' visits. The existing situation—the lack of places where both visitors and patients could spend some time in safety, constitutes a problem for the GUHA that needs to be resolved.

Furthermore, the administration should also identify if something went wrong concerning the design of the premises, such as amenities for the facilitation of people with disabilities, in order to re-design them promptly and be able to provide the appropriate solutions.

With the aim of the hierarchical cluster analysis, it was proven that the employees evaluate the GUHA as an integrated space in terms of a workplace. In particular, they relate comfort features with the infrastructures' usage and the spatial architecture design.

Respectively, and according to the findings of two-step cluster analysis, the medical staff show the highest satisfaction with the healthcare unit environment and its spaces. In contrast, the lowest satisfaction was expressed by the nursing or other hospital staff. As regards the administrative staff, it is satisfied mainly by the comfort and spatial design of the healthcare unit spaces. The opposite concept depicts their views on the GUHA usage infrastructures. More specifically, job satisfaction is associated with satisfaction they derive by their income and workplace conditions. It became apparent that the nursing and other staff are the ones that undergo the more stressful conditions. This category of hospital employees is in the first line concerning the direct contacts with patients and their relatives. Direct contact multiplies the possibility to experience workplace violence such as verbal attacks [41], while nurses undergo severe occupational injuries, such as back injuries, due to the very vigilant handing of patients [2]. Thus, GUHA administration should place on the top of its agenda specific measures to improve stress relief solutions, especially for this group of its staff.

**Author Contributions:** P.K. and V.A. designed the research and the methods and materials for its implementation. S.T. designed the questionnaire used for the data collection and supervised the project progress. A.T. conducted the fieldwork and data collection. P.K., S.T. and V.A. conducted the data analysis and wrote the paper in close cooperation. All authors have read and agreed to the published version of the manuscript.

**Funding:** This research received no external funding.

**Conflicts of Interest:** The authors declare no conflict of interest.

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
