# Peer review of "Indoor and Outdoor Design in Healthcare Environments: The Employees’ Views in the General University Hospital of Alexandroupolis, Greece"

_environments, doi:10.3390/environments7080061_

Round 1

Reviewer 1 Report

Thanks a lot for conducting this study regarding an important topic yet to be explored. My major concern is about the scope of the study which is considered so huge and have been investigated extensively; despite making it specifically related to Greece will make it more valuable, the authors could still borrow many thoughts and scientific concepts from other studies that have been conducted rigorously in the US. Here are few suggestions:

Introduction: can be improved by exploring some of the recent publications (after 2015) about the same topic in Health Environment Research and Design Journal. Adding related studies will enhance the quality of the study.

Method: The methodology section could be improved if the authors could borrow ideas from the existing tools to explore the same question. Then, the related ideas about Greece can be fit into the study.

Results: I do not have so much to say about the results and analysis since it can be improved but maybe better than other sections.

Discussion: It is expected to tie the results into the existing findings; so please compare the findings with the the revised scholar articles and their findings.

Conclusion: It is highly suggested to consider and generalize the findings to the context in Greece. 

Author Response

Response to Reviewer’s comments_ Reviewer 1

Comments and Suggestions for Authors

Thanks a lot for conducting this study regarding an important topic yet to be explored. My major concern is about the scope of the study which is considered so huge and have been investigated extensively; despite making it specifically related to Greece will make it more valuable, the authors could still borrow many thoughts and scientific concepts from other studies that have been conducted rigorously in the US. Here are few suggestions:

  • Introduction: can be improved by exploring some of the recent publications (after 2015) about the same topic in Health Environment Research and Design Journal. Adding related studies will enhance the quality of the study. – The introduction was enriched with recent publications of the requested area
  • Method: The methodology section could be improved if the authors could borrow ideas from the existing tools to explore the same question. Then, the related ideas about Greece can be fit into the study. There are very interesting proposals we can adapt from papers from the same topic in Health Environment Research and Design Journal for improvement in the introduction or in the discussion. The method was applied at the research is impossible to change at this phase.
  • Results: I do not have so much to say about the results and analysis since it can be improved but maybe better than other sections. - This section has been improved
  • Discussion: It is expected to tie the results into the existing findings; so please compare the findings with the revised scholar articles and their findings. New bibliography was added in order to compare the findings of the study with other findings
  • Conclusion: It is highly suggested to consider and generalize the findings to the context in Greece. - Findings represent a health unit in Greece while it is expected that they also address other areas with similar characteristics.

Finally, the manuscript was assigned to a native speaker for overall proof reading.

Reviewer 2 Report

The subject is highly topical.

The research plan is coherent and adequately structured. Questionnaires and measurement processes are well designed and reliable. 

Statistic analysis and evaluation results are particularly interesting. 

Conclusions are supported by the results, and would become useful in future design strategies.

Tables provide an accurate perspective of each factor.

Figure 1 is too small, and should be enlarged. 

Author Response

Response to Reviewer’s comments_ Reviewer 2

Comments and Suggestions for Authors

The subject is highly topical. The research plan is coherent and adequately structured. Questionnaires and measurement processes are well designed and reliable. Statistic analysis and evaluation results are particularly interesting. Conclusions are supported by the results, and would become useful in future design strategies. Tables provide an accurate perspective of each factor.

Figure 1 is too small, and should be enlarged. Figure 1 was enlarged as requested

Finally, the manuscript was assigned to a native speaker for overall proof reading.

Reviewer 3 Report

This paper addressed an important and increasingly relevant topic about the role of physical environments for health and well-being. It aims to understand the impact of the hospital environment on staff at the General University Hospital of Alexandroupolis, in Greece.

The overall aims and topic are of merit and relevance to the readership. But the paper currently falls well short of the standard and quality one might expect for publication.

The authors might want to consider:

  1. The introduction lacks specificity and conceptual clarity. There needs to be a more up to date and theoretically informed overview of literature on the role of indoor and outdoor green space in hospital contexts. For example see Ulrich 2008 (https://pubmed.ncbi.nlm.nih.gov/21161908/)
  2. The aim of the study needs to be more clearly signposted in the introduction with detail on the question to be addressed (with reference to what kind of staff; what type of environments; what outcomes).
  3. The methods claims that interviews are the best way to collect statistical data. This is plainly not true and I wonder if the authors meant a survey as this label is also used. There needs to be far more detail about the range and type of data collected with justification for the variables collected.
  4. I don't follow the so-called random allocation of surveys. Why did the authors not try to maximise the sample by giving it to all eligible staff? Also what were the eligibility criteria?
  5. The use of factor analysis is not well justified or explained nor is the analysis plan coherent. What data were to be analysed and why?
  6. The scope and content of the green space improvements need to be defined and characterised as without any sense of what features are being evaluated it is difficult to judge the results or compare with other studies.
  7. The results need to be more clearly reported and presented - the tables and figures don't add value to the narrative which is confused and seemingly out of place with a study that is trying to asses the relationship between the environment and psychological stress. I don't know if indeed that is the aim and whether the outcome is psychological stress, making it impossible to discern the significance of the results.
  8. The conclusion needs to put the results in context to comparable studies about hospital design and health outcomes. Here linking the study to a defined theoretical framework drawing on environmental psychology and biophillic design principles would help contextualise the placement of this study. 

Author Response

Response to Reviewer’s comments_ Reviewer 3

Comments and Suggestions for Authors

This paper addressed an important and increasingly relevant topic about the role of physical environments for health and well-being. It aims to understand the impact of the hospital environment on staff at the General University Hospital of Alexandroupolis, in Greece. The overall aims and topic are of merit and relevance to the readership. But the paper currently falls well short of the standard and quality one might expect for publication. Τhe authors might want to consider:

  1. The introduction lacks specificity and conceptual clarity. There needs to be a more up to date and theoretically informed overview of literature on the role of indoor and outdoor green space in hospital contexts. For example see Ulrich 2008 (https://pubmed.ncbi.nlm.nih.gov/21161908/) – the suggested bibliography has been added and the literature was reviewed with new relative information
  1. The aim of the study needs to be more clearly signposted in the introduction with detail on the question to be addressed (with reference to what kind of staff; what type of environments; what outcomes). – The aim was reformatted and enriched with the type of the personnel and environments and the major findings emerged: “The case study was focused on the evaluation and improvement of the premises comprising the General University Hospital of Alexandroupolis, Greece. To this end, it was attempted to investigate and analyze - from a sociological aspect - the employees’ views and correlations concerning the impact of the hospital design, and particularly addressing indoor and outdoor spaces. In line with the existing settings, direct and indirect contact with the natural environment was also examined in this study. Indicative results reveal that there is a deficiency in green spaces, although the hospital staff recognizes their beneficial role in improving the hospital as a workplace. Specific solutions were proposed addressing interior and landscape design eligible for healthcare environments with similar characteristics, such as the creation of a garden, the placement of natural flowers or plants, and encouragement of taking care of them. The suggested guidelines are based on bioclimatic architecture, landscape architecture, the existing climate and environmental conditions and also by taking into consideration that nursing staff falls to high levels of occupational stress.
  1. The methods claims that interviews are the best way to collect statistical data. This is plainly not true and I wonder if the authors meant a survey as this label is also used. There needs to be far more detail about the range and type of data collected with justification for the variables collected. - The comment of the reviewer is very accurate; the interviews are one way to collect data. All the collected variables described and analyzed to the results
  2. I don't follow the so-called random allocation of surveys. Why did the authors not try to maximise the sample by giving it to all eligible staff? Also, what were the eligibility criteria? - The comment of the reviewer is very repartee. Researches indifferent professional spaces to be carried out many problems should be solved. First, we must have the permission on terms to be determined by the director of the hospital (for example not creating problems to the operation of the hospital and in limit of time). In the most of our researches we used simple random sampling to determine the sample of the people we ask to the survey. In the specific case if we used the same method, we should have a list of names of the sample and try to approach them to complete the questionnaire. Also, we have to persuade the people that the survey is anonymous. To avoid this kind of reactions we choose to approach the personnel with a very friendly way to give them the questionnaire and tell them to put it in a box when they complete it. This was happened only during their break time so the operation of the hospital did not avoid. Also, the method of the total inventory was not possible to be used because of the very long time needed so the limitation of the director of the hospital was forbidden.
  3. The use of factor analysis is not well justified or explained nor is the analysis plan coherent. What data were to be analysed and why? - In our research we used factor analysis two times. First time to evaluate the outdoor environment of the hospital and the second to evaluate the indoor environment. Factor analysis was applied when we have a multivariable (question) that consisting from many variables in order to group them and take better interpretation to our data. To the factors resulted from the factor analysis, hierarchical cluster analysis was applied to corelate the factors indoor and outdoor environment
  4. The scope and content of the green space improvements need to be defined and characterised as without any sense of what features are being evaluated it is difficult to judge the results or compare with other studies. - We approached the issue from a sociological aspect in order to examine what do they employees consider and how they relate the indoor and outdoor environment of the hospital. At the second level, what you suggest is an interesting proposal for future research on how the hospital space is structured. The first that we have focused on in this research involves the perception of space while the second - the architecture of space. The architect creates spaces, for instance by taking into account the functionality of spaces (the distance between the doctor’s office and the consultation room where he visits his/her patients ), the lighting, etc. In our research we ask the doctor - employee how he/she feels when he works in an existing space. Few things can change in this space - in fact, in the conclusion of the manuscript, some suggestions are made for the improvement of the spaces. However, we regard that this one as well as similar studies could be used by architects to improve the spaces they design (taking into account new factors such as how children will be kept occupied in hospitals in order to avoid unpleasant situations and relative problems).
  5. The results need to be more clearly reported and presented - the tables and figures don't add value to the narrative which is confused and seemingly out of place with a study that is trying to assess the relationship between the environment and psychological stress. I don't know if indeed that is the aim and whether the outcome is psychological stress, making it impossible to discern the significance of the results. – In the same way that animals have their own space in which they move and perform, so do humans need their space to perform their activities. The study investigates how employees perceive their workplace. We do not delve into the psychology of employees. Yet we reach the level of how employees feel in this area and how it is possible to make them feel better with small interventions such as the placement of flower pots. As mentioned before, there are other methods of approaching the issue from a different point of view.
  6. The conclusion needs to put the results in context to comparable studies about hospital design and health outcomes. Here linking the study to a defined theoretical framework drawing on environmental psychology and biophillic design principles would help contextualise the placement of this study. – this section has been improved with the introduction of new related outcomes in order to support the study findings. In conjunction with the general improvement of the introduction and discussion has met the requested improvements.

Finally, the manuscript was assigned to a native speaker for overall proof reading.

Round 2

Reviewer 3 Report

Thank you to the authors for attempting to revise the manuscript in line with the first round of peer review.

I find the paper still rather confusing and am rather mystified by the introduction of a 'sociological' frame of reference with no prior mention of sociological theory - how is this study sociological?

The structure of the paper is slightly improved but there are still problems such as stating preliminary results at the end of the introduction. 

I don't know what bioclimatic architecture is and this is not explained. 

The authors seem to have missed my point about interviews and statistical data and the manuscript still maintains that interviews are a common method for collecting statistical data. 

Sampling is not explained nor is the use of factor analysis explained - which is critical to understanding the methods used (and this is approach is not common in sociology). 

Author Response

Response to Reviewer’s comments_ Reviewer 3_2nd Round of review

Comments and Suggestions for Authors

  1. I find the paper still rather confusing and am rather mystified by the introduction of a 'sociological' frame of reference with no prior mention of sociological theory - how is this study sociological?

The case study surveys the perception of space. In order for this approach to be more comprehensible please let us give an example:

We invite two students to our office and place the three guest chairs we have in our office next to each other, at an equal distance from us. We will notice that the students will sit in the two remote chairs leaving space between them. If the students sit next to each other it means that they either have a very friendly relationship or they are a couple. If we investigate the reasons that each student behaves in this way, we enter the field of psychology. If we investigate how we should place (spatial planning) the chairs to get the different behaviors of students this is architectural design (for instance, the positioning of benches in parks leads to two different uses: namely the use of those who want contact with other people and the use of those who seek isolation). If we ask the users of the benches - on the two different “use types” and correlate it with some other parameters - variables then we perform a sociological research. This is what we do in the case study. If we aim to examine how the citizens (society) behave in relation to space, then we enter the science of sociology. In this case, sociologists must have developed some theories which will we take as granted in order to conduct our research. Unfortunately, such a theory has not been developed by sociologists so far. There few studies on architectural design of green spaces, addressing interior and exterior spaces in houses and some others including the citizens’ perceptions on the way how they conceptualize the mean space like the case study.  We share the opinion that our study cannot reject a theory that does not exist; whereas, it  creates the conditions to trigger discussion on the so called “perception of space”.

  1. The structure of the paper is slightly improved but there are still problems such as stating preliminary results at the end of the introduction.

At the end of the Introduction we are not stating the preliminary results of the paper, but we describe the aim of the manuscript as happens in most research articles.

  1. I don't know what bioclimatic architecture is and this is not explained.

Two references were included in the manuscript to give a useful definition of Bioclimatic design and further explain the terminology.

  1. Watson D. Bioclimatic Design. In Sustainable Built Environments, Loftness, V., Haase, D., Εds.; Springer: New York, USA, 2013; pp.1-30. https://doi.org/10.1007/978-1-4614-5828-9 
  2. Tovar Alcázar, M.R., García Chávez, J.R. Educational Program for Promoting the Application of Bioclimatic and Sustainable Architecture in Elementary Schools. Energy Procedia 2014, 57, 999-1004. https://doi.org/10.1016/j.egypro.2014.10.083
  1. The authors seem to have missed my point about interviews and statistical data and the manuscript still maintains that interviews are a common method for collecting statistical data. Sampling is not explained nor is the use of factor analysis explained - which is critical to understanding the methods used (and this is approach is not common in sociology).

In the most of our studies we used simple random sampling to determine the sample of the questioned people we address to the surveys. In the specific case study if we used the same method, we should have a list of the employees’ names comprising the sample and try to approach them to complete the questionnaire. Also, we should have persuaded these people that the survey was anonymous. To avoid this kind of reactions we have chosen to approach the personnel under a very friendly manner - to give them the questionnaire and tell them to put it in a box after its completion. The procedure took place strictly during their breaks, so the operation of the hospital was not disturbed. Also, the method of the total inventory was not possible to be used because of the very long time needed so the limitation of the director of the hospital was forbidden.

Factor analysis is applied when we have a multivariable (question) consisting from many variables, in order to group them, and get better interpretations of our data. An example constitutes a recent research at the same journal that applies similarly methods of factor and hierarchical analysis:

 Fanelli R-M. 2020.  The Spatial and Temporal Variability of the Effects of Agricultural Practices on the Environment. Environments 2020, 7(4), 33; https://doi.org/10.3390/environments7040033

In our research we used factor analysis two times. First time to evaluate the outdoor environment of the hospital and the second one to evaluate the indoor environment To the factors resulted from the factor analysis, hierarchical cluster analysis was applied to corelate the factors indoor and outdoor environment.

Also, we should emphasize to the reviewer, as we have already mentioned in the first round of review, that our research is a sociological research - addressed to citizens (society). We would like also to clarify that it is not sociology and we do not use sociological methods. Yet we consider that a sociologist will be motivated by this research and its findings in order to utilize them and to formulate his/her sociοlogical theories.

*Please note that the manuscript was send for proof reading by a native speaker

Round 3

Reviewer 3 Report

The authors have offered an explanation about bioclimatic design and provided a rationale for factor analysis which were the two major issues. I still struggle with the use of sociological as a descriptor of the nature of the research but will leave others to make sense of this.